# Real world relevance of generative counterfactual explanations

Swami Sankaranarayanan[1], Thomas Hartvigsen[1], Lauren Oakden-Rayner[2], Marzyeh Ghassemi[1], and
Phillip Isola[1]

[1]Massachusetts Institute of Technology, USA
[2]University of Adelaide, South Australia, Australia

## Abstract

The interpretability of deep learning based algorithms is critical in settings where
the algorithm must provide actionable information such as clinical diagnoses or
instructions in autonomous driving. Image based explanations or feature attribu-
tions are an often-proposed solution for natural imaging datasets, but their utility
for mission critical settings is unclear. In this work, we provide image explana-
tions that are both semantically interpretable and assess their utility for real world
relevance using imaging data extracted from clinical settings. We address the
problem of pneumonia classification from Chest X-ray images where we show that
(1) by perturbing specific latent dimensions of a GAN based model, the classifier
predictions can be flipped and (2) the latent factors have clinical relevance. We
demonstrate the latter by performing a case study with a board-certified radiologist
and identify some latent factors that are clinically informative and others that may
capture spurious correlations.

## 1 Introduction

Imagine your doctor orders a CT scan, and it indicates an abnormality. Naturally, you would ask
your doctor "Why?". An effective explanation should be both spatially localized to the abnormality
discovered by the CT scan and be easily interpreted by a non-expert. Regardless of who examines
the scan, be it an algorithm or a doctor, a successful explanation must satisfy these criteria to be
trustworthy [Lipton, 2018]. Effectively explaining complex predictions is critical for follow-up tasks,
like deciding how to treat your discovered abnormality. However, while explanations are crucial to
real-world impact, existing ML systems remain largely opaque. To continue expanding ML systems
into impactful domains, we need explanation systems that aid real world practitioners [Doshi-Velez
and Kim, 2017].

Explainable machine learning is a rapidly-growing research area [Arrieta et al., 2020]. A recent
and promising direction is to explain model predictions using counterfactuals learned by generative
models [Wachter et al., 2017, Singla et al., 2019, 2021, Lang et al., 2021]. However, while recent
works appear to provide meaningful explanations of model predictions, there has been limited work
demonstrating whether experts actually find the explanations helpful in practice [Van Calster et al.,
2019]. We take a first step towards stress-testing generative models in real, expert-driven applications.

Our approach is outlined in Figure 1 for explaining an off-the-shelf pneumonia classifier. We start by
training a generative model on the domain of interest. In order to explain the prediction of a classifier
on a real image, we project the image into the latent space of the generative model. We find that
perturbing specific latent dimensions induces a localized change in the image which in turn affects
the classifier prediction. We conduct a clinical review with a domain expert to obtain the real world
relevance of the chosen latent dimensions.

2022 Trustworthy and Socially Responsible Machine Learning (TSRML 2022) co-located with NeurIPS 2022.

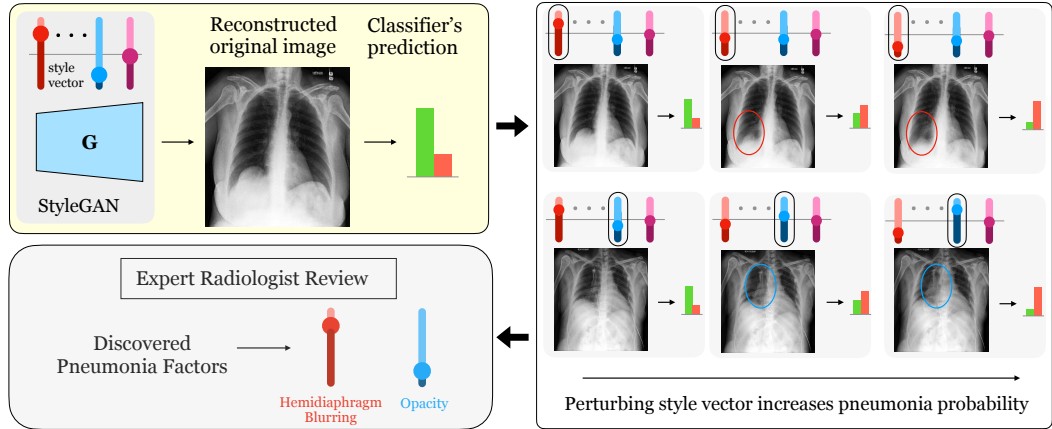

Figure 1: **Explaining pneumonia classifier**: We sample an image that is scored as negative for pneumonia by the classifier and attempt to identify the *factors* that flip the classifier's decision. As a first step, we project the input image to the latent space of the styleGAN using an optimization procedure, which provides us access to the latent style vector corresponding to the input. Several dimensions of this style vector, when perturbed in the latent space, modifies meaningful regions of the input image which in turn flip the classifier decision. We employ an expert-in-the-loop setting to identify the clinical relevance of these dimensions.

In summary, our contribution is two-fold:

- We introduce an expert-in-the-loop explanation method that audits the predictions of a pre-trained classifier using counterfactuals derived from a StyleGAN. Our approach allows us to identify semantically-meaningful *factors* that have large impacts on the classifier's predictions.

- We then showcase our approach by auditing a Pneumonia classifier trained on Chest X-Ray images. By collaborating with a radiologist, we ultimately find that our method successfully identifies clinically-meaningful factors that convince the model to flip its predictions.

## 2  Method

The objective of our approach to explain the decision of an off-the-shelf classifier that was trained to predict a pathology from an image inputs. Specifically, let $C : X \rightarrow y$ represent the pre-trained classifier that takes an image $X \in \mathbb{R}^{H \times W}$ as input and predicts the relevant pathology probability $y$, where $y \in [0, 1]$. The classifier is calibrated such that 0.5 is the decision threshold for a positive prediction.

In order to obtain attributes that *meaningfully* affect the decision of C, we need a generative model of the image manifold that captures the set of factors that can potentially affect the classifier. For this purpose, we train a StyleGAN-2 Karras et al. [2020] based generator that has been shown to contain disentangled attributes in its style-based latent space Wu et al. [2021].

Given a pretrained generative model, one can synthesize fake images by randomly sampling from the latent space of the generator. Thus the true latent vector for a fake image is known. However, for a real image $X$, the true latent vector is unknown. Finding the latent vector for a real image or *inverting the GAN* is an active area of research Xia et al. [2021]. Existing methods either use an optimization based procedure to backproject a given instance into the latent space of a pretrained generative model or train an encoder model whose output is the estimate of the true latent vector. In this work, we *invert* the GAN using backprojection as it does not require external training data and it is better suited for imposing domain specific priors.

## 2.1 Obtaining latent vectors for real images

Given an input image $X$, the backprojection procedure involves performing gradient descent over the space of the latent vectors optimizing a loss function that encodes a combination of pixel-wise constraints and domain specific constraints. Specifically:

$$z^* = \text{argmin}_z \mathcal{L}(G(z), X) \tag{1}$$

The contents of the loss function $\mathcal{L}$ are as follows:

- Mean square loss ($L_{mse}$): Enforces similarity at the pixel level.
- Perceptual loss ($L_{pips}$): Enforces perceptual similarity over features across multiple scales.
- Domain specific loss ($L_{cls}$): Enforces domain similarity by utilizing a pretrained classifier trained on the same domain as $X$.
- Discriminator loss ($L_{disc}$): Adversarial loss term using the pretrained discriminator that ensures that the resultant image is as realistic as possible.

In summary, the full loss function being optimized is:

$$\mathcal{L} = L_{mse} + \alpha L_{pips} + \beta L_{cls} + \gamma L_{disc} \tag{2}$$

Note that, each loss term is applied directly on the pixel space and not on the latent space. During the optimization procedure, we start from initializing $z$ to the mean latent vector computed over 100k images randomly sampled from $G$. Similar to previous works that perform backprojection, we find that this offers increased stability to the inversion process. For each image, 500 gradient descent steps are performed to obtain the estimated latent vector.

While the backprojection procedure described in Eq. 2 can be used for inverting any GAN based generator, specific details for a styleGAN based generator need to be accounted for. As explained in the StyleSpace work [Wu et al., 2021], different latent spaces of styleGAN offer tradeoffs for inversion fidelity against editability. Following their suggestion, we invert the latent style space, as it offers a better trade-off for inversion and editability. This is an important concern for our approach, since we need to ensure that the resultant latent vector: (1) offers high fidelity for inversion, so that we do not miss any critical detail in medical images and (2) remains in distribution for the generator even after applying a perturbation.

## 2.2 Identifying dimensions that change classifier's decision

In the previous section, we outlined the approach for inverting the image $X$ to the GAN latent space. This operation can be represented as a projection function: $P : X \rightarrow$ s.t $G(S) \cong X$, where $S \in \mathbb{R}^D$ is the D-dimensional latent vector, commonly referred to as style vector Wu et al. [2021]. The optimization function defined in Eq 2. In this section, we describe our approach to identify specific latent dimensions that when perturbed, affect the prediction of an underlying classifier, that is trained using data from domain $\mathcal{X}$.

Given an image $X$, let $S = [s_1, s_2, ..., s_{d-1}, s_d, s_{d+1}, ..s_D]$ be the latent style vector that generated $X$. Let's denote the classifier score for the original image as $C(X; S)$ making the dependence on $S$ explicit. To assess the effectiveness of the latent dimension $d$, we perturb the style value at dimension $d$ by clamping it to maximum and minimum values and record the change. We estimate the minimum and maximum values for each dimension of the latent vector over a set of randomly sampled images from the latent space of the GAN generator. Let $S^{d+}$ denote the perturbed style vector such that the $d^{th}$ dimension is replaced by the maximum: $S^{d+} = [s_1, s_2, ..., s_{d-1}, s_d^{max}, s_{d+1}, ..s_D]$ and corresponding for $S^{d-}$.

Starting from the classifier score on the original input, we obtain the following set of perturbed classifier scores and the corresponding difference scores, represented by $\triangle$:

$$C_X^d = \{ C(X; S^{d-}), C(X; S), C(X; S^{d+}) \} \tag{3}$$

$$\triangle_X^{d-} = C(X; S) - C(X; S^{d-}) \tag{4}$$

$$\triangle_X^{d+} = C(X; S^{d+}) - C(X) \tag{5}$$

In order to quantify the effect of perturbing this dimension on the classifier, we evaluate the classifier with perturbed inputs corresponding to the extreme values for each dimension $\{s_d^{min}, s_d^{max}\}$. In this procedure, we clamp each dimension of $S$ *separately* and *independently* from others to the min and max values for that corresponding dimension.

We aggregate the classifier deltas across a dataset of heldout images and select dimensions whose delta values across the entire set exceed a preset value $\delta$, which we treat as a hyperparameter.

## 3  Case study: what factors affect an off-the-shelf pathology classifier?

We perform a small-scale case study whose objective is to identify the latent factors that can explain the predictions made by an off-the-shelf pathology classifier. Specifically, we choose the problem of pneumonia classification from Chest X-ray (CXR) images and use an off-the-shelf classifier that has been shown to be highly performant on this problem [Cohen et al., 2020]. The classifier was trained on a combination of several CXR datasets [Wang et al., 2017, Johnson et al., 2019, Irvin et al., 2019] using a multi-label training objective consisting of 18 pathologies. We only perturb the classifier score for pneumonia in our experiments. The clinical annotations used in this work were provided by a board-certified radiologist, who is also a co-author of this work.

### 3.1  Experimental Setup

**Dataset and Pretraining** For our experiments, we use the publicly available CheXpert dataset [Irvin et al., 2019] that consists of 200k Chest X-ray (CXR) images. To train the StyleGAN model, we filter the dataset to contain only *frontal AP* scans. We split the dataset into training and test splits with the training split containing 80% of the images. The generative model is trained for a total of 120 epochs (or 20k kimgs), early stopping the model based on FID and Precision-Recall scores.

**Latent Factor Selection** We use our approach described in Section 2.2 to select the latent factors that affect the predictions of our pneumonia classifier. We perform the selection on the held out test split that consist of 2000 images. We use a classifier delta threshold of $\delta = 0.3$. As a result of our selection process, we end up with a total of 10 dimensions (out of 7424 latent dimensions) that change the classifier score beyond $\delta$.

**Setup for Clinical Review** For each identified latent factor, we sort the held out set in the decreasing order of effected change and choose the top 5 images where the magnitude of the change is the largest. This results in a set of 50 images over 10 chosen latent factors. In order to minimize any bias due to a domain shift, we only use those image where the classifier is extremely confident in its prediction i.e. images where the classifier predicts the correct label with a confidence of 0.9 or more. We also ensure to pick images where the GAN reconstruction is very accurate and does not remove any detail in the clinically relevant spatial regions.

### 3.2  Identifying clinically relevant factors

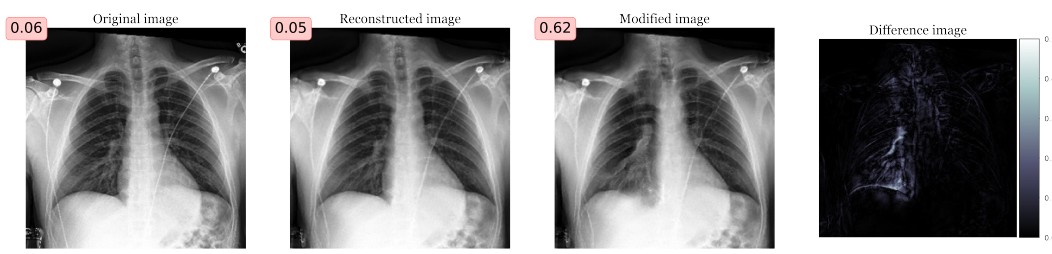

Figure 2: **Sample input for Clinical review**

We conduct a small-scale clinical review by an clinical radiologist in order to semantically annotate the latent factors that change the classifier's predictions. During the review, the expert is shown the input images and the modified images resulting from the perturbation process. A sample input is shown in Figure 2. For each image, we include the pneumonia probability output of the off-the-shelf classifier that we are attempting to explain. They are also presented with a gif with the modified

image stacked on top of the unmodified image in order to better visualize the difference. They are then presented with the following queries for each input:

- Is the change semantically meaningful? If yes, please provide a clinical description of this change.
- Is the change relevant to the pathology under consideration i.e. pneumonia?

The clinically relevant latent factors identified as a result of the review procedure are demonstrated in Figure 3. The review data including the responses of the expert are presented in the Appendix 5. From the limited number of images (50) provided to our expert across 10 latent factors, they were able to ascertain 4 factors whose perturbations induced clinically meaningful changes in the image and increased the classifier score for pneumonia. In addition, out of the 50 instances, 23 were found to contain perturbations that were relevant to the pathology under consideration i.e. pneumonia.

In addition to being clinically relevant, these factors are able to *explain* the behavior of the classifier. As Figure 3 demonstrates, the classifier score is susceptible to these identified factors such as *increased opacity in the right medial base* or *increased density in the right heart border* etc. From the point of view of critical applications such as healthcare, our explanation provides an interactive way to explain the classifier behavior as opposed to more passive explanation provided by approaches based on feature attributions or spatial relevance alone.

It is worth noting that the generative model was provided no knowledge of the classifier or any other pathology based prior during the training procedure. The pneumonia classifier was used as a domain based prior only when performing the image reconstruction and the latent factor selection steps (Section 2.2). The existence of clinically relevant latent dimensions in the latent space of a pretrained StyleGAN generative model implies that these dimensions might be generalizable to other downstream tasks within the same domain as well.

### 3.3 Caveats of our approach

**Are all discovered latent dimensions useful?** While some dimensions we discovered did lead to relevant behaviors there were others that were harder to interpret. Out of the 10 dimensions that were reviewed by our expert, 4 of them presented with pathology related changes. For most examples where any of the other 6 dimensions were perturbed, the changes were more global – perhaps related to spurious correlations like image quality rather than pathology-related image features. A promising direction for future work would be to explore interesting behavior among the latent style dimensions that are possible spurious correlations.

**Should generator know about the classifier?** The knowledge of the classifier was provided during the reconstruction stage, not during the training of the generator – the objective is to make the generator become a domain expert. Infusing knowledge of a specific pathology at the training stage would require training multiple generators for different pathologies. This knowledge is provided this knowledge implicitly when we use the generator to reconstruct a real x-ray image via a classifier based loss. We have made this distinction more explicit.

**Comparisons against other approaches** This work showcases a set of promising initial results making a case for counterfactual explanations. As future work, there needs to be a broader study comparing the clinical relevance of counterfactuals (vs) a popular explanation approach, such as salience maps. Our experiments would follow previous such studies for saliency maps Saporta et al. [2022].

## 4 Related work

The popular approaches used to explain image classifiers are based on spatial attributions such as heatmaps or saliency maps [Selvaraju et al., 2017, Rebuffi et al., 2020] or feature attributions Xu et al. [2020]. Such explanations highlight the regions of the image that are helpful for classifier prediction or visualize concepts modeled by neurons in deep networks. Counterfactual explanations Wachter et al. [2017] have emerged as a promising explanation mechanism for back box image classification models especially in cases where the attributes that are used by the classifier are not distinct objects present in the image. Counterfactual explanations work by providing an alternative set of inputs where only some attributes are changed while the others are held constant, thus helping isolate

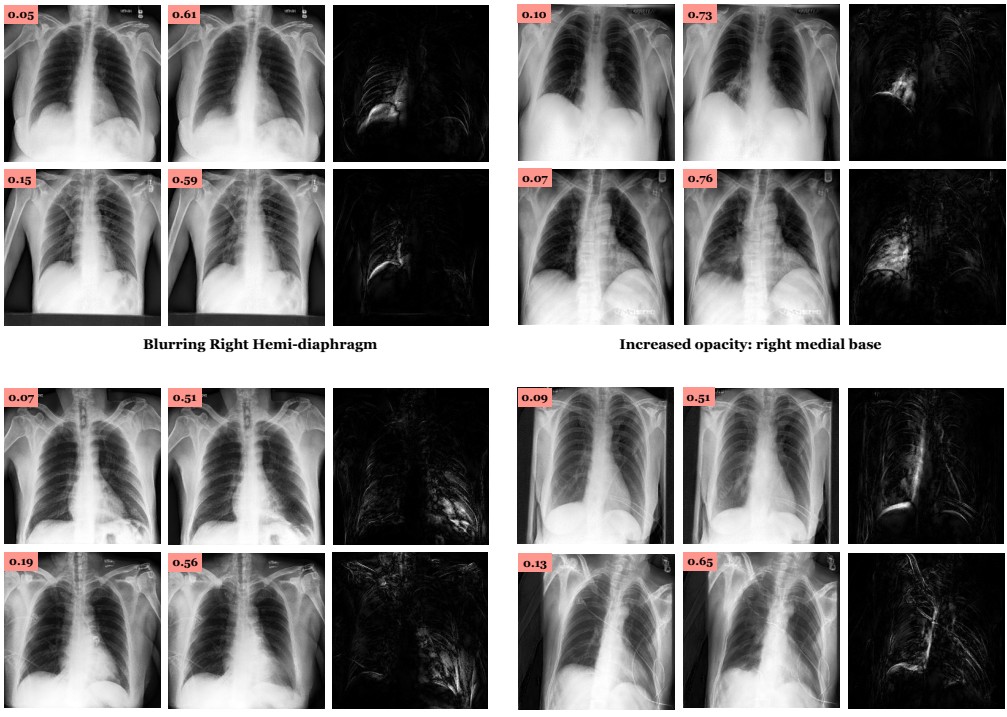

**Blurring Right Hemi-diaphragm**    **Increased opacity: right medial base**

**Increase left basal opacity**    **Increased density: right heart border**

Figure 3: **Semantic factors identified by Clinical expert**: This figure shows the different semantic factors identified during the Clinical review procedure. In each row, we show the image as reconstructed by the generative model, its modified version, resulting from the style-vector based perturbation approach detailed in Section 2.2 and a difference image. At the top left of each image, we show the prediction of the classifier for pneumonia. As illustrated, the factors mined from the latent space of the pretrained GAN have clinical meaning and change the classifier scores significantly.

how a classifier is affected by a specific set of attributes. Recent approaches for image based counterfactuals [Singla et al., 2019, 2021] have extensively used generative models due to their ability to provide a disentangled latent space Wu et al. [2021] where each attribute can be controlled separately. The work closest to our proposed approach is StylEx Lang et al. [2021], which obtains style based explanations from the latent space of a pretrained generative model. The key difference in our work is the application of a generative model based explanation approach to a critical data domain such as healthcare including an expert-in-the-loop review. While StylEx can be applied successfully to domains where semantic attributes are easily describable, such as faces, animals etc, real world applications require more transparency and interpretability, which is the key contribution of our framework.

## 5    Conclusion and Future work

Machine learning models are notorious for being black boxes and the larger such models are, the harder it becomes to understand their decisions. In this work, we take a first step towards attributing real world meaning for classifier explanations. We demonstrated our approach on a high impact problem of pneumonia risk computation from Chest X-ray images. We found semantically meaningful dimensions within the latent space of a GAN model. We showed that (1) perturbing those dimensions can create counterfactual inputs that flip the pneumonia classifier decision and (2) the latent dimensions have clinical relevance by performing a review with a clinical expert. While our initial results are promising, there are several avenues for improvement including robustness of our latent factor selection procedure, improving our GAN model based reconstruction of real images and performing a more rigorous large scale case study with multiple clinical experts across different pathologies. We leave these as avenues for future work.

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

## Appendix

## A   Data from Clinical Review

The radiologist was presented with 50 instances with each instance consisting of the original, reconstructed and perturbed images stacked side by side as shown in Figure 2. We summarized our findings from this review in Section 3.2. Here, we present the response from our expert for the clinically relevant dimensions:

Table 1: Clinical review responses for the relevant style dimensions.

| Style dimension | Image index | Clinical Relevance | Pneumonia relevance |
|---|---|---|---|
| **1980** | 1 | slight increase in density along right heart border | possible |
| | 2 | increased density along right heart border and blurring of the border | early change of pneumonia |
| | 3 | right retrocardiac density and right diaphragm blurring | possible, not compelling |
| | 4 | again some right heart border and hemidiaphragm blurring | not compelling |
| | 5 | blurring right heart border | subtle but possible |
| **2743** | 6 | blurring right heart border and hemidiaphragm | early change of pneumonia |
| | 7 | blurring right heart border and hemidiaphragm | early change of pneumonia |
| | 8 | minimal change | no |
| | 9 | blurring right heart border and hemidiaphragm | early change of pneumonia |
| | 10 | blurring right hemidiaphrag | can be, but unnatural change |
| **3499** | 11 | hazy opacity RML and blurring right heart border | yes |
| | 12 | right heart border and hemidiaphgram blurring with increased opacity | yes, subtle |
| | 13 | minimal change | no |
| | 14 | increased opacity right medial base | yes, but probably wouldn't diagnose |
| | 15 | increased opacity right medial base | yes, but probably wouldn't diagnose |
| **3526** | 16 | left basal increased opacity | yes, but would probably not even notice |
| | 17 | blurring diaphragms | not in isolation without opacity as well |
| | 18 | left basal increased opacity | relevant but not enough alone |
| | 19 | left basal increased opacity | yes |
| | 20 | minimal change | no |

