# OpenReview forum: "Real world relevance of generative counterfactual explanations"
_NeurIPS.cc/2022/Workshop/TSRML — TSRML2022_

### Official Review · Reviewer_iyst · 2022-10-14
**Interesting setup, but are the results really encouraging? Could be an interesting work to discuss**

**Overall Rating:** 7

**Summary:**

The paper considers the generation of counterfactual examples for lung radiographs, to explain a classifier for pneumonia.
- The authors propose a setup where a StyleGAN learns to generate counterfactual images (e.g., that exhibit pneumonia for patients who do not have it) based on changing a few latent dimensions.
- The authors investigate whether the images generated by varying the top-10-salient latent dimensions appear to be reasonable to clinicians and whether clinicians can assign a clinically-meaningful label to what has changed in the image.
- The authors obtain what I interpret to be *mixed results*. At least the paper shows that, *in principle*, their setup is able to generate clinically-meaningful counterfactual radiographs.



**Strengths:**

- The paper involves actual clinicians looking at an actual problem, IMHO this is very important to foster future advancements in XAI
- The proposed method is interesting: the StyleGAN "*can*" generate counterfactual images that explain a flip in the prediction based on changes in the image that are considered to be clinically-meaningful
- The paper is written very clearly, it has a linear story and puts focus on important details.
- IMHO the proposal of seeking meaningful explanations is an interesting point of discussion, irrespective of whether the method is successful.

**Weaknesses:**

- It is not clear what the message of this paper is: is this supposed to be a success story or the opposite? In particular, it appears that finding *some* clinically-meaningful explanations required a lot of work: Latent Factor Selection (Sec.3.1) is applied on the *test set*, latent factors are *ranked* to top-10, only images for which the classifier is very confident (>= 0.9) are considered. *Even then*, 4 out of 10 factors appear to lead to clinically-meaningful distortions, for roughly half of the images.
- Can/should one trust a classifier "more" if these sorts of examples *can* be achieved? What if this classifier is still very much subject to break under e.g. image artifacts? These important questions are not really answered here. What is then the merit of seeking clinically-meaningful explanations in particular over others?
- The authors consider cases in which the classifier is very confident about its prediction (>=0.9) but, arguably, one most needs XAI when the classifier is not.



**Overall Recommendation:**

I believe that this paper is well executed and the proposed approach is interesting.
Albeit the outcome of the study is not clear in terms of how promising this really is (to me, it looks like it is not), I believe it could be fertile ground for discussion either way.
This is especially because it is not clear to me whether it makes sense to seek such sort of *intuitive* (e.g., as done here, clinically meaningful) explanations, or whether we should instead seek all sorts of explanations, including non-intuitive ones (e.g., to decide that the classifier should *not* be trusted).

**Review Confidence:**

4: The reviewer is confident but not absolutely certain that the evaluation is correct

---

### Official Review · Reviewer_89UV · 2022-10-18
**Review of Paper102 (lean towards accept)**

**Overall Rating:** 6

**Summary:**

The authors propose a way to explain an image classifier by learning a projection function that maps the original image to some latent space of a generative model. They then implement their approach using an off-the-shelf pneumonia classifier and present the learned latent factors to a radiologist for inspection. The authors find that a subset of these latent factors are verified by the radiologist to be meaningful based on the radiologist's domain knowledge.

**Strengths:**

This work did a good job in terms of motivating their work with a real problem and then conducting an evaluation of the proposed method with real data and real users (which is rarely the case with ML work). Props to the authors for actually performing an evaluation with a radiologist.

**Weaknesses:**

I am a bit confused by the motivation for using a generative model to explain a pre-trained classifier. Specifically, in L153, where the authors say that the generative model "was provided no knowledge of the classifier". If the goal is to explain the classifier, then isn't it counterintuitive to try an approach that has no knowledge of the classifier it is trying to explain? While the ability of the proposed method to identify domain-relevant factors is interesting, it feels disconnected from the primary motivation in the introduction.

Other comments:
1. I am unsure if the method is novel. While Section 2.2 seems important in terms of the proposed method, it was actually the portion of the paper that was most difficult to follow. Specifically, for someone unfamiliar with styleGAN notation, $S^D$ and other similar terms are not defined so I am unsure if these are supposed to be images, latent vectors, or something else.

2. Lack of evaluation against other methods. The authors claim that their method is a way to improve the interpretability of a pre-trained classifier. However, as they acknowledge, there exist many other explanation approaches for image-based networks. I would have liked to see the radiologist also consider the usefulness of these other methods against the new explanations proposed in this work.

**Overall Recommendation:**

I have some concerns about the method itself and the thoroughness of the evaluation but the work falls within the scope of the workshop and presents an interesting evaluation with a human-in-the-loop. For that reason, I would lean towards accepting but recommend authors take into consideration the weaknesses discussed above.

**Review Confidence:**

3: The reviewer is fairly confident that the evaluation is correct

---

### Decision · Program_Chairs · 2022-10-23

**Decision:**

Accept

**Comment:**

Very nice work on connecting counterfactual explanations to real world scenarios in radiographs.